# The Gut Microbiome in Stevens–Johnson Syndrome and Sjögren’s Disease: Correlations with Dry Eye

**DOI:** 10.3390/microorganisms13122730

**Published:** 2025-11-29

**Authors:** Luciana Frizon, Talita Trevizani Rocchetti, André Frizon, Rafael Jorge Alves de Alcântara, Cintia S. de Paiva, José Álvaro Pereira Gomes

**Affiliations:** 1Department of Ophthalmology and Visual Sciences, Escola Paulista de Medicina, Universidade Federal de São Paulo, Botucatu Street 822, São Paulo 04023-062, Brazil; talita.rocchetti@gmail.com (T.T.R.); andre.frizon@gmail.com (A.F.); rja.alcantara@gmail.com (R.J.A.d.A.); japgomes@uol.com.br (J.Á.P.G.); 2Department of Ophthalmology, Baylor College of Medicine, Baylor Plz, Houston, TX 77030, USA; cintiadp@bcm.edu

**Keywords:** Stevens–Johnson syndrome, Sjögren’s disease, gut microbiome, dry eye

## Abstract

Changes in gut microbial composition may influence mucosal immune responses and contribute to systemic autoimmune manifestations. In this pilot exploratory study, we investigated and compared the gut microbiome in patients with Stevens–Johnson syndrome (SJS), patients with Sjögren’s disease (SjD), and healthy controls, using next-generation sequencing (NGS), and assessed correlations with dry eye parameters. The study included 10 patients with SJS matched by age and sex to 10 healthy controls, and 10 patients with SjD matched to an additional set of 10 healthy controls. Dry eye parameters were employed to evaluate dry eye disease (DED). Microbiome profiles were determined using next-generation sequencing of the 16S V3-V4 region and analyzed using the Silva database. The gut microbiome exhibited significant differences in the SJS group, including a reduced Chao1 index (*p* = 0.01) that was progressively correlated with increased ocular severity and a decrease in *Faecalibacterium* (*p* = 0.048) compared to the healthy control group. In the SJS group, strong correlations were observed between increased *Christensenellaceae* with decreased DED DEWS (Dry Eye Workshop score) (*p* = 0.04), increased *Subdoligranulum* with decreased NEI (National Eye Institute) score (*p* = 0.04), and increased *Clostridia* and longer TBUT (tear break-up time) (*p* = 0.009). In contrast, the gut microbiome of SjD patients was similar to that of healthy controls. Patients with SJS exhibited distinct alterations in gut microbial composition, characterized by reduced microbial richness and depletion of *Faecalibacterium*. Furthermore, a significant association was found between specific bacterial taxa and milder dry eye severity, suggesting a possible link between changes in the gut microbiome and inflammation of the ocular surface.

## 1. Introduction

The human microbiome, comprising trillions of microorganisms distributed across various body sites, plays a crucial role in development, immune regulation, and maintaining a physiological balance. The gut hosts the most abundant and diverse microbial population, which is shaped by diet, lifestyle, and environment [1,2]. Dysbiosis, marked by reduced microbial richness and diversity, can impair systemic homeostasis and has been linked to diseases affecting distant organs, including the eyes [1,2]. The concept of a “gut–eye axis” has emerged, implicating gut microbial imbalances in ocular inflammatory and autoimmune disorders, such as Sjögren’s disease (SjD), dry eye disease (DED), uveitis, and diabetic retinopathy [3,4,5,6].

Stevens–Johnson syndrome (SJS) and SjD are systemic autoimmune diseases with severe ocular manifestations, including persistent dry eye, that are often refractory to conventional therapies. SJS is a rare condition, with an annual prevalence of 1.2 to 6 cases per million, characterized by acute mucocutaneous inflammation and chronic cicatricial sequelae, which lead to irreversible damage to the ocular surface and eyelids, severely impairing vision [7,8]. Genetic and immune-mediated mechanisms, particularly T cell-mediated type IV hypersensitivity reactions and activation of Toll-like receptor 3, play a critical role in the recurrent mucocutaneous inflammation and severe ocular complications observed in SJS [8,9]. SjD, in contrast, primarily affects exocrine glands, contributing to dryness of the eyes and mouth, as well as systemic inflammatory manifestations. It is more prevalent, affecting 100 to 900 per million annually, with a strong female predominance [10,11,12]. Both diseases share clinical features, including severe dry eye, chronic conjunctival inflammation, and reduced salivary flow [7,10].

DED, common in both conditions, is characterized by tear film instability, hyperosmolarity, inflammation, and neurosensory abnormalities, creating a self-perpetuating inflammatory cycle [13,14]. Recent studies suggest that the gut microbiota modulates T-cell responses and contributes to ocular surface homeostasis. Animal and human models have demonstrated that alterations in the gut microbiota can exacerbate dry eye severity via immune dysregulation [15,16,17].

While gut microbiome alterations have been explored in SjD [15,18,19], and ocular surface dysbiosis has been reported in SJS [20,21,22,23], no study has examined the intestinal microbiome in SJS to date. This study is the first to characterize the gut microbiome in SJS patients and compare it with that of individuals with SjD and healthy controls while also assessing correlations with dry eye severity. We hypothesize that SJS is associated with intestinal dysbiosis, which may potentially contribute to ocular surface inflammation and the clinical severity of dry eye disease.

## 2. Materials and Methods

### 2.1. Study Design and Patient Selection

This pilot exploratory, prospective, and transversal study was approved by the Ethics Committee of the Federal University of São Paulo (approval number: 6.003.698) and conducted following the Declaration of Helsinki. All participants provided written informed consent before participation.

Participants were recruited from the Corneal and External Diseases Clinic at the Federal University of São Paulo between March 2023 and February 2024. Two independent comparison sets were analyzed: (1) ten patients with SJS matched by age and sex to ten healthy controls, and (2) ten patients with SjD matched by age and sex to a distinct group of ten healthy controls. Distinct control groups were created for each condition due to differences in age and sex distribution between the SJS and SjD patient cohorts. One participant in the SJS group was excluded after a pregnancy was identified subsequent to sample collection and microbiome processing. Therefore, the final SJS cohort included 9 patients. Participants were aged eighteen years or older, had no other ocular diseases (other than those related to SJS or dry eye), and had no history of gastrointestinal disorders. Exclusion criteria included pregnancy, infection, or surgery in the last three months, as well as ocular or systemic antibiotic and probiotic use during that period.

The SJS group included patients with chronic disease following a history of mucocutaneous inflammation induced by medications or infections, involving at least two mucous membranes. The classification into mild, moderate, and severe ocular involvement was based on the presence and extent of ocular surface sequelae, following the simplified criteria proposed initially by Sotozono et al. [24] and later adopted by Kittipibul et al. [21] (Table 1). These included eyelash abnormalities, lid margin keratinization, conjunctival inflammation, conjunctival fibrosis, limbal stem cell deficiency, corneal epitheliopathy, and corneal opacity. Patients presenting fewer or milder findings were classified as mild, those with moderate structural involvement as moderate, and those with multiple and severe complications as severe.

The SjD group consisted of patients with confirmed primary SjD, diagnosed according to the American College of Rheumatology/European League Against Rheumatism criteria [25] with no associated autoimmune comorbidities. All SjD patients reported systemic involvement and symptoms of dry eye and dry mouth, and all were receiving oral hydroxychloroquine (HCQ) at a dose of 400 mg/day, with a treatment duration of two years or longer. Three patients were receiving chronic systemic immunosuppressant therapy (methotrexate, azathioprine, or leflunomide), and one patient was using an oral corticosteroid (prednisone 20 mg). A detailed summary of all topical and systemic medications used by participants in the SJS and SjD groups is provided in the Appendix A.

Healthy control subjects had no eye irritation, a tear break-up time (TBUT) ≥ 7 s, Schirmer I ≥ 10 mm, corneal fluorescein score ≤ 2, conjunctival lissamine score ≤ 2, and no meibomian gland disease. Subjects were excluded if they had prior laser-assisted in situ keratomileusis or corneal transplantation surgery, cataract surgery in the past year, punctal occlusion with plugs or cautery, a history of contact lens wear, use of topical medications other than preservative-free artificial tears, or chronic use of systemic medications known to reduce tear production.

### 2.2. Clinical Assessment

All participants underwent a comprehensive ophthalmologic evaluation performed by the same examiner (LF). Ocular disease involvement was assessed according to the guidelines established by the International Dry Eye Workshop (DEWS) [26]. The examination was conducted in a standardized sequence: initially, the Ocular Surface Disease Index (OSDI) questionnaire was administered, followed by the Schirmer I test, tear break-up time (TBUT), and finally, corneal fluorescein and conjunctival lissamine green staining, graded according to the National Eye Institute (NEI) scoring system. Dry eye severity was classified using the DED DEWS system [13], which ranges from 0 to 4 and is based on the cumulative assessment of clinical parameters, including OSDI, Schirmer I test, TBUT, NEI score, and subjective symptoms. Accordingly, more advanced stages of dry eye are characterized by higher DED DEWS, OSDI, and NEI scores and lower TBUT and Schirmer I values. All SJS and SjD patients had a DED DEWS ≥ 3, and the healthy control group had a score = 0.

The Schirmer I test assessed basal and reflex tear secretion over a five-minute period without the use of anesthetic eye drops [27]. TBUT was measured three times in each eye using fluorescein strips, and the average was recorded. Corneal fluorescein staining was scored across five regions (0–3 per region), with a maximum score of 15. Conjunctival staining with lissamine green was assessed over six areas, with scores from 0 to 18. The NEI score was the sum of both corneal and conjunctival staining [28].

### 2.3. Sample Collection

All participants received standardized stool microbiome collection kits with detailed instructions for at-home sampling. They were advised to collect the fecal sample within 48 h before the clinical visit, store it at ambient temperature, and return it on the day of the clinical examination. All samples were collected using standardized fecal microbiome collection kits (DNA/RNA Shield^TM^ Zymo Research Corp, Irvine, CA, USA), which contain a preservation medium capable of maintaining microbial integrity at ambient temperature for up to 30 days [29]. After clinical examination, samples were stored at ambient temperature and shipped to the laboratory (Inside Diagnósticos, São Paulo, SP, Brazil) within 48 h. Upon arrival, all samples were processed for sequencing within 2 days.

### 2.4. Fecal Microbiome Analysis

DNA extraction was performed using the ZymoBIOMICS™ DNA Kit (Zymo Research Corp., Irvine, CA, USA), following the manufacturer’s protocol. Amplicon sequencing of the 16S rRNA V3–V4 region was subsequently carried out using the QIAseq 16S/ITS Screening Panel (Qiagen GmbH, Hilden, Germany), according to the manufacturer’s instructions. The primers 515F (5′-GTGCCAGCMGCCGCGGTAA-3′) and 806R (5′-GGACTACHVGGGTWTCTAAT-3′) were employed for amplification. Sequencing was carried out on the Illumina MiSeq platform using the MiSeq v2 kit (Illumina, San Diego, CA, USA). Raw sequencing reads were processed using QIIME 2 version 2024.5 [30]. Default parameters were applied for trimming and joining paired-end reads. The DADA2 plugin was used to denoise reads and generate amplicon sequence variants (ASVs), which were subsequently clustered into operational taxonomic units (OTUs) at 99% similarity using the VSEARCH plugin. Taxonomic assignment was performed based on the SILVA database (release 138.2), with taxonomic filtering to exclude mitochondria, chloroplasts, and Eukaryotic taxa [31]. The sequencing depth ranged from 45,423 to 308,901 sequences. The median value was 113,211 (Q1 = 71,588; Q3 = 204,054). Given the high sequencing depth, the samples were not rarefied in order to preserve rare sequence data in the analyses [32].

### 2.5. Statistical Analysis

Statistical analysis was conducted using SPSS (version 20.0, SPSS Inc., Chicago, IL, USA) for general statistical tests and RStudio (version 4.3.1) for microbiome-specific analyses. Mean comparisons between the two groups were evaluated using Student’s *t*-test, Fisher’s exact test, and the Mann–Whitney test. Comparison among more than two groups, analysis of variance (ANOVA), and Kruskal–Wallis tests were used. Spearman’s correlation was applied to assess linear associations between variables due to non-normal distributions [33]. In the genus-level analysis, only genera with an average relative abundance of at least 1% were included. Statistical significance was set at *p* = 0.05.

Microbiome data analysis and visualization were performed using the phyloseq, vegan, ggplot2, and microbiome packages in RStudio. The alpha diversity, Chao1, and Shannon indices were calculated, and statistical significance was assessed using the Mann–Whitney test with Dunn’s adjustment for multiple comparisons. Beta diversity was computed using both unweighted and weighted UniFrac distances and visualized through principal coordinate analysis (PCoA) plots. Statistical significance for beta diversity was determined using permutational multivariate analysis of variance (PERMANOVA) with the adonis and pairwise functions.

## 3. Results

### 3.1. Clinical Data

Age and sex distribution exhibited no significant differences between the SJS group and the respective healthy control group, and between the SjD group and the respective healthy control group (Table 2). The clinical examination and dry eye parameters of the SJS group and its control group, as well as the SjD group and its control group, are described in Table 3. Patients with SJS and SjD exhibited significantly worse clinical indicators, including higher OSDI and NEI scores and lower Schirmer I and TBUT values, compared to their respective healthy controls. The composite DED DEWS, which integrates all of these parameters, was also markedly higher in both disease groups (3–4), reflecting greater ocular surface involvement. All comparisons between the disease group and its control group identified statistically significant differences (*p* < 0.005) in dry eye parameters (Table 3).

### 3.2. Intestinal Microbiome Measures

Concerning alpha diversity, the Chao1 index, which represents species richness, was significantly lower in the SJS group compared to the respective healthy control group (*p* = 0.012) and demonstrated a progressive decline correlating with increased ocular surface severity (*p* = 0.01) (Figure 1A,C). In contrast, the Shannon diversity index did not show a significant difference between groups (*p* = 0.175) (Figure 1B). Alpha diversity comparisons between the SjD group and the respective healthy control group indicated numerical differences in the Chao1 and Shannon diversity indices, but these were not statistically significant (*p* = 0.181 and *p* = 0.377, respectively) (Figure 1A,B). Beta diversity analysis using weighted and unweighted UniFrac distances revealed no significant differences between the SJS group and its respective healthy control group (*p* = 0.192) or between the SjD group and its respective healthy control group (*p* = 0.757) (Figure 1D,E).

Table 4 shows the phylum and genus abundance of the SJS group and the respective healthy control group, and Figure 2 illustrates the phylum and genus abundance of both SJS and SjD and their respective healthy control groups. Among phyla, no statistically significant differences were observed between SJS and SjD when compared to their respective healthy control groups (Figure 2A). Regarding genus, *Faecalibacterium* was significantly less abundant in the SJS group compared to the respective healthy control group (*p* = 0.048). This result was further supported by the Hedges’ g estimate (Hedges’ g = 0.897), indicating a strong association between SJS and reduced *Faecalibacterium* abundance. (Figure 2B,C and Table 4). No significant differences in genus levels were found between the SjD group and the respective healthy control group (Figure 2B). Raw sequencing data and OTU tables have been deposited in the NCBI public repository and can be accessed at URL https://www.ncbi.nlm.nih.gov/bioproject/1280506 (accessed on 21 June 2025).

### 3.3. Dry Eye Correlations

Comparing phyla data with dry eye indices in the SJS group, Spearman’s correlation analysis revealed significant negative correlations between the NEI Score and Actinobacteriota (r = −0.695, *p* = 0.039) as well as Synergistota (r = −0.722, *p* = 0.028) (Figure 3A,B). These findings suggest that higher abundances of Actinobacteriota and Synergistota were associated with less severe dry eye symptoms, such as reduced corneal and conjunctival staining in the SJS group. In the SjD group, no statistically significant correlations were observed at the phylum level or with respect to dry eye indices.

In correlations between bacterial genera and dry eye indices in the SJS group, Spearman’s correlation revealed a moderate negative correlation between *Christensenellaceae* abundance and the DED DEWS (r = −0.696, *p* = 0.037), as well as between *Subdoligranulum* abundance and the NEI score (r = −0.690, *p* = 0.039) (Figure 3C,D). A positive correlation was also observed between *Clostridia* abundance and TBUT (r = 0.803, *p* = 0.009) (Figure 3E). These results indicate that higher abundances of *Christensenellaceae*, *Subdoligranulum*, and *Clostridia* are associated with less severe dry eye parameters in the SJS group. In the SjD group, a moderate negative correlation was found between *Alistipes* abundance and the DED DEWS (r = −0.684, *p* = 0.029), suggesting that a higher abundance of *Alistipes* was associated with less severe dry eye disease (Figure 3F). When applying FDR correction using the Benjamini–Hochberg method, only the correlation between *Clostridia* abundance and TBUT (*p* = 0.045) remained statistically significant (Appendix A).

Regarding the ocular SJS group severity (mild, moderate, and severe ocular severity grades), Spearman’s correlation analysis revealed a moderate positive correlation between disease severity and the abundance of Cyanobacteria (*p* = 0.050, r = 0.659) and Fusobacteria (*p* = 0.050, r = 0.659) at the phylum level. No statistically significant correlations were observed at the genus level.

Taken together, SJS patients exhibited reduced gut microbial richness, as indicated by a lower Chao1 index and decreased abundance of *Faecalibacterium*, with specific bacterial taxa correlating with reduced dry eye severity.

## 4. Discussion

The results of this study show that patients with SJS exhibited significantly reduced gut microbial richness, including marked depletion of *Faecalibacterium*, compared to healthy controls. Moreover, distinct alterations in microbial composition were associated with milder dry eye parameters. To our knowledge, this is the first investigation to characterize the gut microbiome in SJS while simultaneously comparing it with patients diagnosed with primary SjD and healthy individuals.

Alpha diversity, particularly the Chao1 index, was significantly decreased in SJS patients and showed a progressive decline, possibly correlating with the severity of ocular surface inflammation. These results are consistent with prior findings in mucous membrane pemphigoid [34], a mucosal autoimmune disease that shares clinical features with SJS, including cicatrizing conjunctivitis and T cell-mediated inflammation. Similar patterns of reduced alpha diversity have been documented in uveitis [35], Behçet’s disease [36], Sjögren’s disease [15,37,38], and diabetic retinopathy [39], reinforcing the hypothesis that gut dysbiosis may play a role in immune-driven ocular pathology via systemic pathways. Reduced microbial diversity is commonly associated with increased systemic inflammation and compromised intestinal barrier function, promoting the expansion of pathogenic taxa and chronic immune activation [12,15].

A prominent finding in our cohort of SJS patients was the depletion of *Faecalibacterium*, a key butyrate-producing genus with well-documented anti-inflammatory properties [6]. This is consistent with reductions reported in uveitis [35], SjD [15,19], Behçet’s disease [36], and mucous membrane pemphigoid [34]. *Faecalibacterium* contributes to intestinal homeostasis by supporting epithelial barrier integrity, inducing tolerogenic dendritic cells and FOXP3+ regulatory T cells (Tregs), and regulating cytokine profiles. Its primary metabolite, butyrate, a short-chain fatty acid (SCFA) produced via microbial fermentation of dietary fiber, plays an essential immunomodulatory role [6]. Schaefer et al. [16,40] demonstrated that intestinal commensals, such as *Faecalibacterium*, can promote ocular immune tolerance by inducing Tregs in draining lymph nodes. We additionally found that the pro-inflammatory genus *Prevotella* was relatively more abundant in the SJS group; however, this increase was not statistically significant.

Consistent with clinical expectations, SJS patients exhibited worse dry eye outcomes, with the DED DEWS ranging from 3 to 4, and microbial diversity inversely associated with ocular surface damage. Correlations between microbiome composition and specific dry eye parameters have previously been identified in SjD [15,18]. Our study further correlated microbial composition at the phylum and genus levels with specific dry eye parameters. At the phylum level, higher abundances of Actinobacteriota and Synergistota were associated with lower NEI scores, suggesting reduced ocular surface damage. Similar findings were reported by Moon et al. [18], linking *Actinobacteriota* with improved TBUT and reduced dry eye severity, and Wang et al. [41] found an association between *Actinobacteriota* and reduced dry eye severity in Sjögren’s disease. Actinobacteria include several commensal genera with recognized anti-inflammatory and epithelial-stabilizing functions, which may contribute to improved mucosal homeostasis [18]. Although less studied, Synergistota has been linked to metabolic and immunomodulatory pathways that may influence epithelial integrity, suggesting a possible indirect protective contribution within the gut–eye axis.

At the genus level, we found that *Subdoligranulum* abundance was negatively correlated with NEI scores, suggesting a potentially protective role against dry eye severity. This is consistent with studies by Cao et al. [42] and Moon et al. [18], who reported depletion of *Subdoligranulum* in both SjD and non-Sjögren dry eye, supporting its association with a healthier gut environment through butyrate production. Increased abundance of *Christensenellaceae* was also correlated with a lower DED DEWS. Members of this family are considered beneficial commensals linked to anti-inflammatory effects, improved epithelial integrity, and lower systemic inflammation [43]. Finally, *Clostridia***,** another SCFA-producing group, was positively associated with TBUT, aligning with evidence that *Clostridia* enhances regulatory immune responses and mucosal barrier function [44]. Together, these taxa may support ocular surface homeostasis through gut-derived immunomodulatory mechanisms relevant to the gut–eye axis.

In our study, alpha- and beta-diversity in the SjD group did not differ significantly from those of healthy controls, in agreement with findings reported by Mendez et al. [19], Moon et al. [18], and Zhang et al. [45], but in contrast to the decreased microbial diversity described by Schaefer et al. [17], de Paiva et al. [15], and Cano-Ortiz et al. [38]. Similarly, microbial composition at both the phylum and genus levels showed no statistically significant differences in our cohort. However, a recent study by Jia et al. [46] using shotgun metagenomic sequencing in treatment-naïve primary SjD revealed marked compositional and functional aberrations in the gut microbiota, characterized by reduced microbial richness along with enrichment of potentially pro-inflammatory taxa.

An important consideration in interpreting the microbiome findings of the SjD group is that all patients were under oral hydroxychloroquine (HCQ) treatment for two years or longer. HCQ has been shown to modulate gut microbial composition in autoimmune diseases, partially reversing dysbiosis and reducing pro-inflammatory bacterial signatures in patients with primary Sjögren’s syndrome [47]. Reviews of gut microbiota–drug interactions also indicate that HCQ can alter overall microbial diversity, shift the abundance of key commensal taxa, and influence host immune responses [48]. Experimental models further support these effects, demonstrating that HCQ reduces diversity and induces measurable changes in microbial composition [49,50]. Thus, the relative similarity observed between the SjD and healthy control groups in our study may reflect treatment-related modulation of the microbiome rather than an absence of disease-associated alterations. Hydroxychloroquine should therefore be considered an important confounding factor, underscoring the need for future studies including medication-naïve patients or stratified analyses.

The use of additional systemic medications as immunosuppressants in the SjD group (such as methotrexate, azathioprine, leflunomide, and oral corticosteroids) may have modulated the gut microbial composition and attenuated disease-related differences. Likewise, the use of topical anti-inflammatory therapies in SJS patients, including corticosteroids and tacrolimus, may have improved ocular surface signs and reduced the strength of the microbiome–clinical associations. These factors represent potential confounders and should be considered when interpreting our findings.

Despite its strengths, this study has several limitations. It is an exploratory pilot investigation with a cross-sectional, single-center design and a relatively small cohort, particularly for a rare condition such as SJS, for which patient recruitment is inherently constrained. Dietary habits were not assessed, which represents an important uncontrolled confounding factor given the strong influence of diet on gut microbial composition. The absence of longitudinal follow-up further limits the ability to draw temporal or causal inferences. Moreover, the 16S rRNA sequencing approach used provides genus-level resolution and does not allow species-level or functional gene-level characterization. Therefore, the findings should be interpreted as exploratory and hypothesis-generating rather than confirmatory. Finally, because the study population was drawn from a single center, the results may not be generalizable to other populations with different genetic backgrounds, dietary patterns, or environmental exposures.

Future studies using shotgun metagenomics, strain-resolved profiling, and integration of host immune and metabolomic data, including assessment of SCFA synthesis pathways and immunomodulatory gene content, are warranted to enhance mechanistic insights.

## 5. Conclusions

In summary, our findings indicate that patients with SJS exhibit reduced gut microbial richness, including a depletion of anti-inflammatory taxa such as *Faecalibacterium*. Moreover, specific taxonomic groups were correlated with milder dry eye parameters. These results provide preliminary evidence supporting the potential role of the gut microbiome in ocular surface inflammation. Although these associations should be interpreted with caution, given the cross-sectional design, small sample size, and interindividual variability, this study provides novel insights into the gut–eye axis and expands the growing understanding of microbiome involvement in ocular immunopathology and SJS-related dry eye. These findings highlight the need for larger, longitudinal studies with adequate statistical power and robust effect-size estimation, ideally integrating functional metagenomics and immunological profiling to validate these associations and to advance microbiome-based therapeutic strategies for SJS-associated dry eye.

## Figures and Tables

**Figure 1 microorganisms-13-02730-f001:**
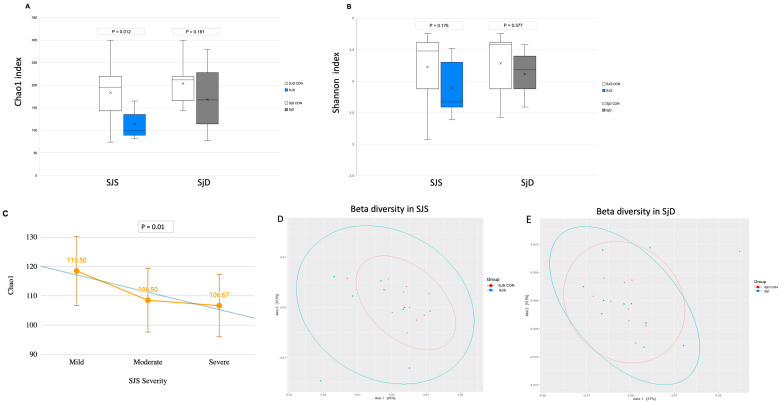
Species richness (Chao1), Shannon index, and beta diversity. (**A**) Chao1 index showed significant differences only between the SJS controls and the SJS group (*p* = 0.012), but not between the SjD controls and the SjD group (*p* = 0.181). (**B**) Shannon diversity index showed no statistical differences in either group (*p* = 0.175 and *p* = 0.377). (**C**) Chao1 index showed a progressive depletion with increased ocular severity in SJS patients. (**D**) Principal coordinate analysis (PCoA) based on weighted UniFrac distances revealed no significant differences in beta diversity between SJS CON and the SJS group (*p* = 0.192) or (**E**) between the SjD CON and SjD group (*p* = 0.757). SJS, Stevens–Johnson syndrome; SjD, Sjögren’s disease; CON, healthy control group.

**Figure 2 microorganisms-13-02730-f002:**
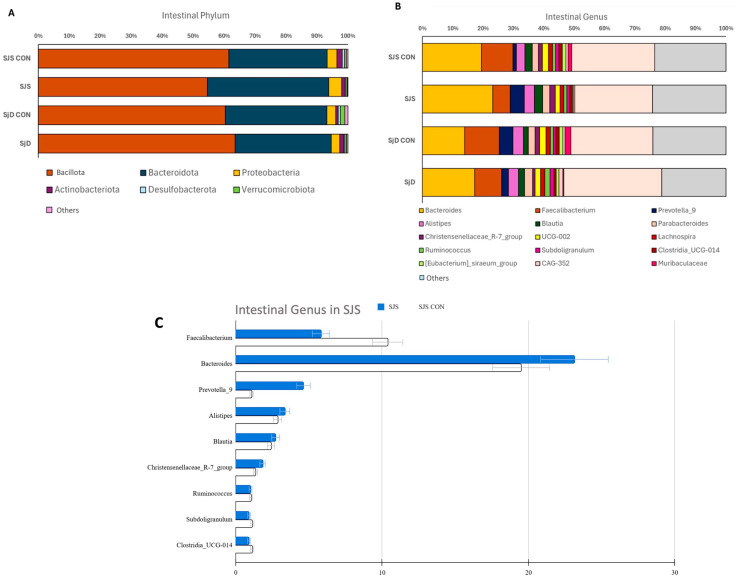
Phylum and genus abundance. (**A**) No statistically significant differences were observed at the phylum level in either group. (**B**,**C**) At the genus level, *Faecalibacterium* was significantly less abundant in the SJS group compared to the SJS CON (*p* = 0.048). *Prevotella* showed higher relative abundance in the SJS group, although without statistical significance. SJS, Stevens–Johnson syndrome; SjD, Sjögren’s disease; CON, healthy control group.

**Figure 3 microorganisms-13-02730-f003:**
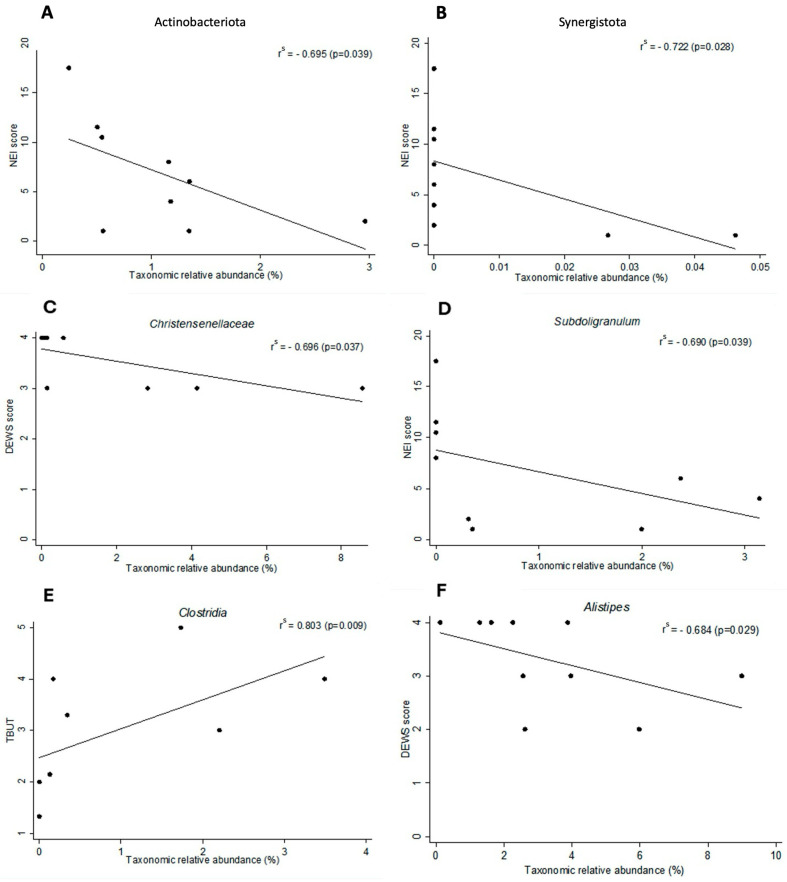
Spearman’s correlation without adjustment between dry eye indices and intestinal microbiome. (**A**,**B**) In SJS, the phyla Actinobacteriota and Synergistota revealed significant negative correlations with NEI Score (*r* = −0.695, *p* = 0.038 and *r* = −0.722, *p* = 0.028, respectively). (**C**) The *Christensenellaceae* revealed a significant negative correlation with DED DEWS (*r* = −0.696, *p* = 0.037), as well as *Subdoligranulum* abundance (**D**) and the NEI Score (*r* = −0.690, *p* = 0.039). (**E**) A positive correlation was also observed between *Clostridia* abundance and TBUT (*r* = 0.803, *p* = 0.009). (**F**) In the SjD group, a moderate negative correlation was found between *Alistipes* abundance and the DED DEWS (*r* = −0.684, *p* = 0.029). NEI score, cornea fluorescein, and conjunctival lissamine green dye staining; DEWS, dry eye disease Dry Eye Workshop score; TBUT, tear break-up time.

**Table 1 microorganisms-13-02730-t001:** Classification criteria for SJS severity, simplified from Sotozono et al. [24].

Ocular Feature	Mild	Moderate	Severe
Eyelash abnormalities	Trichiasis or distichiasis involving less than half of the eyelid	Trichiasis or distichiasis involving more than half of the eyelid	Trichiasis or distichiasis involving the entire eyelid
Lid margin keratinization	Continuous keratinization up to one-third of the eyelid	Continuous keratinization from one-third to one-half of the eyelid	Continuous keratinization involving more than half of the eyelid
Conjunctival inflammation	Conjunctival hyperemia up to ++	Conjunctival hyperemia +++	Conjunctival hyperemia ++++
Conjunctival fibrosis	Conjunctival scarring up to the fornix shortening	Symblepharon without restriction of mobility	Symblepharon with restricted ocular motility
Limbal stem cell deficiency	Up to 180°	Between 180° and 270°	More than 270°
Corneal epitheliopathy	Punctate keratopathy and epithelial erosions	Epithelial defect	Epithelial defect with stromal involvement
Corneal opacity	Mild haze	Moderate haze	Diffuse haze obscuring anterior chamber details

The + sign indicates the clinical intensity of ocular surface alterations. Grading scale: ++ moderate, +++ marked, and ++++ severe alterations.

**Table 2 microorganisms-13-02730-t002:** Demographic characteristics of study groups.

	N, Subjects	Age, Mean, Years	Age, Range, Years	Female/Male
SJS controls	10	40	18–54	8/2
SJS	9	37	24–65	6/3
*p* value		0.517 ^a^		0.628 ^b^
SjD controls	10	49	39–57	10/0
SjD	10	50	44–60	10/0
*p* value		0.492 ^a^		1 ^b^

Comparison sets: SJS (*n* = 9; one exclusion due to pregnancy) matched to 10 healthy controls, and SjD (*n* = 10) matched to 10 healthy controls. *p* values were calculated using Student’s *t*-test ^(a)^ and Fisher’s exact test ^(b)^. SJS, Stevens–Johnson syndrome; SjD, Sjögren’s disease.

**Table 3 microorganisms-13-02730-t003:** Summary of clinical data, showing mean ± standard deviation.

	OSDI	Schirmer I Test	Tear Break-Up Time	NEI	DED DEWS Score
	Score	mm	Seconds	Score	(Number of Patients)
SJS controls	1.03 ± 2.02	32.20 ± 10.16	13.06 ± 2.34	0.25 ± 0.42	0 (10/10)
SJS	48.26 ± 25.71	12.72 ± 13.14	2.98 ± 1.20	6.83 ± 5.60	3 (4/9)
					4 (5/9)
*p* value	*p* < 0.001 ^a^	*p* = 0.002 ^a^	*p* < 0.001 ^a^	*p* = 0.008 ^a^	*p* < 0.001 ^b^
SjD controls	1.23± 2.00	32.00 ± 10.01	12.68 ± 2.26	0.25 ± 0.42	0 (10/10)
SjD	41.13 ± 23.89	10.15 ± 11.78	6.27 ± 3.28	4.20 ± 3.19	3 (4/10)
					4 (6/10)
*p* value	*p* < 0.001 ^a^	*p* < 0.001 ^a^	*p* < 0.001 ^a^	*p* = 0.003 ^a^	*p* < 0.001 ^b^

*p* values were calculated using Student’s *t*-test ^(a)^ and Fisher’s exact test ^(b)^. SJS, Stevens–Johnson syndrome; SjD, Sjögren’s disease; OSDI, Ocular Surface Disease Index questionnaire; NEI score, corneal fluorescein and conjunctival lissamine green dye staining; DED DEWS, dry eye disease Dry Eye Workshop score.

**Table 4 microorganisms-13-02730-t004:** Phylum and genus abundance in SJS and the respective healthy control group.

	Control (N = 10)	SJS (N = 9)	*p* Value
Phylum—Abundance (%), Mean ± SD (Min–Max)			
Actinobacteriota	1.74 ± 1.59 (0.50 to 4.70)	1.09 ± 0.81 (0.24 to 2.96)	0.288
Bacteroidota	31.71 ± 9.85 (20.03 to 54.45)	39.21 ± 13.79 (18.11 to 63.09)	0.191
Campylobacterota	0.00 ± 0.01 (0.00 to 0.02)	0.00 ± 0.00 (0.00 to 0.00)	0.343
Cyanobacteriota	0.29 ± 0.35 (0.00 to 0.84)	0.06 ± 0.13 (0.00 to 0.38)	0.120
Desulfobacterota	0.68 ± 0.70 (0.00 to 2.46)	0.36 ± 0.43 (0.00 to 1.27)	0.305
Elusimicrobiota	0.04 ± 0.13 (0.00 to 0.41)	0.00 ± 0.00 (0.00 to 0.00)	0.343
Euryarchaeota	0.10 ± 0.16 (0.00 to 0.47)	0.05 ± 0.08 (0.00 to 0.18)	0.546
Bacillota	61.57 ± 10.51 (37.34 to 76.27)	54.59 ± 13.45 (34.70 to 75.33)	0.191
Fusobacteriota	0.00 ± 0.00 (0.00 to 0.00)	0.05 ± 0.14 (0.00 to 0.41)	0.126
Proteobacteria	3.14 ± 1.45 (0.93 to 4.97)	4.14 ± 2.34 (1.59 to 8.54)	0.462
Synergistota	0.04 ± 0.10 (0.00 to 0.31)	0.01 ± 0.02 (0.00 to 0.05)	0.636
Thermoplasmatota	0.06 ± 0.11 (0.00 to 0.26)	0.01 ± 0.02 (0.00 to 0.06)	0.252
Verrucomicrobiota	0.63 ± 0.68 (0.00 to 1.97)	0.42 ± 0.63 (0.00 to 1.89)	0.458
Genus—Abundance (%), Mean ± SD (Min–Max)			
*Faecalibacterium*	10.38 ± 5.08 (1.17 to 16.63)	5.81 ± 4.62 (0.00 to 12.65)	0.048
*Prevotella_9*	1.08 ± 3.20 (0.00 to 10.19)	4.63 ± 9.52 (0.00 to 27.49)	0.275
*Alistipes*	2.85 ± 2.27 (0.00 to 7.65)	3.35 ± 2.91 (0.11 to 7.94)	0.806
*Bacteroides*	19.50 ± 13.46 (8.02 to 53.07)	23.14 ± 15.53 (1.16 to 51.24)	0.514
*Blautia*	2.41 ± 1.81 (0.65 to 6.95)	2.73 ± 1.96 (0.00 to 5.59)	0.683
*Christensenellaceae*	1.36 ± 1.63 (0.00 to 4.86)	1.83 ± 2.93 (0.00 to 8.56)	0.870
*Clostridia_UCG-014*	1.12 ± 1.14 (0.00 to 3.04)	0.90 ± 1.27 (0.00 to 3.49)	0.836
*Lachnospira*	1.34 ± 1.34 (0.00 to 3.88)	1.21 ± 1.35 (0.00 to 3.64)	0.901
*Parabacteroides*	1.97 ± 1.26 (0.67 to 4.57)	2.24 ± 1.42 (0.84 to 4.86)	0.462
*Ruminococcus*	1.03 ± 1.21 (0.00 to 2.68)	1.01 ± 1.07 (0.00 to 2.90)	0.933
*Subdoligranulum*	1.13 ± 0.95 (0.00 to 3.13)	0.91 ± 1.24 (0.00 to 3.14)	0.410
*UCG-002*	1.94 ± 2.00 (0.00 to 5.95)	1.64 ± 1.35 (0.00 to 3.26)	0.967

The table presents the mean value comparison between SJS and the respective healthy control group. Statistical significance was evaluated using the Mann–Whitney test. At the genus level, *Faecalibacterium* was significantly less abundant in the SJS group compared to the SJS CON (*p* = 0.048). This finding was further supported by the effect size estimate (Hedges’ g = 0.897), indicating a strong association between SJS and reduced *Faecalibacterium* abundance. SJS, Stevens–Johnson syndrome.

## Data Availability

Publicly available datasets were analyzed in this study. Raw sequencing data and OTU tables have been deposited in the NCBI public repository and can be accessed at URL https://www.ncbi.nlm.nih.gov/bioproject/1280506 (accessed on 21 June 2025).

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
