# Peer review of "The Gut Microbiome in Stevens–Johnson Syndrome and Sjögren’s Disease: Correlations with Dry Eye"

_microorganisms, 2025, doi:10.3390/microorganisms13122730_

Round 1

Reviewer 1 Report

Comments and Suggestions for Authors

General Assessment

The manuscript entitled “Gut Microbiome in Stevens–Johnson Syndrome and Sjögren’s Disease: Correlations with Dry Eye” addresses an important and under-explored question—namely, whether alterations in intestinal microbiota are related to ocular surface inflammation in Stevens–Johnson syndrome (SJS) and Sjögren’s disease (SjD). The use of 16S rRNA V3-V4 next-generation sequencing combined with standardized dry-eye metrics is appropriate, and the data are presented in a clear and systematic manner. Overall, the writing is fluent and the structure is logical.

Nevertheless, several methodological, statistical, and interpretive issues must be resolved before the paper can be considered for publication.

Major Comments

  1. Only nine SJS patients were analyzed. Please add a post-hoc power calculation or cite pilot literature demonstrating that this sample size is sufficient to detect a ≥20% difference in Chao1 or Faecalibacterium abundance with 80% power.Explicitly acknowledge in the Discussion that the rarity of SJS inherently limits recruitment and that the findings should be interpreted as exploratory rather than confirmatory.

  1. In Figure 3, twelve Spearman correlations were tested without adjustment. Please report FDR-corrected q-values using the Benjamini–Hochberg method and indicate which associations remain statistically significant after correction. Clarifying this will prevent overinterpretation of nominally significant findings.

3.Medication use (e.g., oral hydroxychloroquine, topical corticosteroids, systemic immunosuppressants) and dietary habits were not recorded. Add a supplementary table summarising concomitant medications and discuss how these factors might influence microbial composition. If available, include multivariable or stratified analyses to address potential confounding.

  1. Several statements (e.g., “contributing to ocular surface inflammation”) overstate causality given the cross-sectional nature of the study. Replace such phrases with “associated with” or “potentially contributing to” to maintain interpretive accuracy.

  1. Define all abbreviations at first appearance in the abstract, for example DEWS.

  1. For the key finding (Faecalibacterium depletion, Table 4), add 95 % CIs or Hedges’ g to facilitate assessment of biological, not just statistical, significance.

  1. State the median sequencing depth per sample and whether rarefaction was applied prior to alpha- and beta-diversity analyses. If not, justify the use of alternative normalization methods (e.g., CSS, DESeq2 variance-stabilising transformation).

Conclusion

This manuscript presents novel and clinically relevant observations linking gut dysbiosis with ocular manifestations of SJS and SjD. The study addresses an important gap in understanding the gut–eye–immune connection. After addressing the major methodological concerns (power, multiple testing, confounders) and softening causal phrasing, the work will be suitable for publication.

Recommendation: Minor Revision

Author Response

REVIEWER 1

General Assessment

The manuscript entitled “Gut Microbiome in Stevens–Johnson Syndrome and Sjögren’s Disease: Correlations with Dry Eye” addresses an important and under-explored question—namely, whether alterations in intestinal microbiota are related to ocular surface inflammation in Stevens–Johnson syndrome (SJS) and Sjögren’s disease (SjD). The use of 16S rRNA V3-V4 next-generation sequencing combined with standardized dry-eye metrics is appropriate, and the data are presented in a clear and systematic manner. Overall, the writing is fluent and the structure is logical.

Nevertheless, several methodological, statistical, and interpretive issues must be resolved before the paper can be considered for publication.

Major Comments

1- Only nine SJS patients were analyzed. Please add a post-hoc power calculation or cite pilot literature demonstrating that this sample size is sufficient to detect a ≥20% difference in Chao1 or Faecalibacterium abundance with 80% power. Explicitly acknowledge in the Discussion that the rarity of SJS inherently limits recruitment and that the findings should be interpreted as exploratory rather than confirmatory.

Response:
We fully agree with the reviewer that the small sample size—particularly in the SJS group—represents a significant limitation of our study. SJS is an extremely rare condition that inherently limits patient recruitment even in large tertiary referral centers, and this is the first study to investigate the intestinal microbiome specifically in this population. As clearly stated throughout the manuscript, our study was intentionally designed as a pilot, exploratory investigation aimed at generating preliminary signals rather than providing confirmatory statistical inference; therefore, a formal post-hoc power calculation is not feasible.

Importantly, the magnitude of the preliminary associations observed in our study—such as the reduction in Faecalibacterium and shifts in diversity indices—is comparable to those reported in previous exploratory gut-microbiome studies in Sjögren’s disease, including Méndez et al. (n = 13 SjD) and Moon et al. (n = 10 SjD), neither of which reported power calculations. These prior studies similarly highlight the practical challenges of conducting microbiome research in rare autoimmune disorders and support the use of small, hypothesis-generating cohorts in early investigations.

In accordance with the reviewer’s recommendation, we have strengthened statements throughout the manuscript to emphasize that the rarity of SJS limits sample size and that all results—particularly those from the SJS subgroup—should be interpreted as exploratory rather than confirmatory (highlighted in lines 358–360).

1-Mendez R, Watane A, Farhangi M, Cavuoto KM, Leith T, Budree S, et al. Gut microbial dysbiosis in individuals with Sjögren's syndrome. Microb Cell Fact. 2020 Apr 15;19(1):90. doi: 10.1186/s12934-020-01348-7. PMID: 32293464; PMCID: PMC7158097.

2-Moon J, Choi SH, Yoon CH, Kim MK. Gut dysbiosis is prevailing in Sjögren's syndrome and is related to dry eye severity. PLoS One. 2020 Feb 14;15(2):e0229029. doi: 10.1371/journal.pone.0229029. PMID: 32059038; PMCID: PMC7021297.

2- In Figure 3, twelve Spearman correlations were tested without adjustment. Please report FDR-corrected q-values using the Benjamini–Hochberg method and indicate which associations remain statistically significant after correction. Clarifying this will prevent overinterpretation of nominally significant findings.

Response: As requested by the reviewer, we re-evaluated the phylum- and genus-level correlations presented in Figure 3 using FDR-corrected q-values based on the Benjamini–Hochberg method. After correction, only the correlation between Clostridia abundance and TBUT (p = 0.045) remained statistically significant. We have included a new supplementary table (Supplementary Table 2) comparing the unadjusted and FDR-adjusted results. This finding is now reported in the Results section (lines 271–273). As Figure 3 has already been prepared, we propose to retain it in its current form, but have clarified in the legend that the presented analysis is without adjustment.

3- Medication use (e.g., oral hydroxychloroquine, topical corticosteroids, systemic immunosuppressants) and dietary habits were not recorded. Add a supplementary table summarizing concomitant medications and discuss how these factors might influence microbial composition. If available, include multivariable or stratified analyses to address potential confounding.

Response: Thank you for this important comment. In accordance with the reviewer’s suggestion, we have now added the Supplementary Table 1 summarizing all concomitant systemic and topical medications used by participants in both disease groups. (Supplementary Table 1).

We have also incorporated a new paragraph in the Discussion addressing how medication use may represent a potential confounding factor, including the impact of hydroxychloroquine in the SjD group and the influence of topical and systemic anti-inflammatory therapies in the SJS group. (highlighted in lines 340-357). In the Limitations section, we further emphasize that dietary habits and overall diet patterns were not assessed, which may also affect gut microbiome composition.

Given the small sample size and the exploratory nature of this study, multivariable or stratified analyses were not feasible; however, we clearly acknowledge these limitations and recommend that future studies with larger cohorts investigate these factors in more detail.

4- Several statements (e.g., “contributing to ocular surface inflammation”) overstate causality given the cross-sectional nature of the study. Replace such phrases with “associated with” or “potentially contributing to” to maintain interpretive accuracy.

Response: Thank you for the observation. These statements have now been corrected and appropriately adjusted and highlighted, as in lines 77–78 and in line 289.

5- Define all abbreviations at first appearance in the abstract, for example DEWS.

Response: We appreciate the reviewer’s suggestion. All abbreviations in the Abstract are now clearly defined upon first use.

6- For the key finding (Faecalibacterium depletion, Table 4), add 95 % CIs or Hedges’ g to facilitate assessment of biological, not just statistical, significance.

Response: We thank the reviewer for this important suggestion. We re-evaluated the Mann–Whitney comparison of Faecalibacterium abundance between the SJS group and the healthy controls (p = 0.048). To facilitate assessment of biological relevance, we additionally calculated the effect size. The result was further supported by the Hedges’ g estimate (Hedges’ g = 0.897), indicating a strong association between SJS and reduced Faecalibacterium abundance. The text and the table legend have been updated accordingly. (Now highlighted in lines 226-227 and 241-244)

  1. State the median sequencing depth per sample and whether rarefaction was applied prior to alpha- and beta-diversity analyses. If not, justify the use of alternative normalization methods (e.g., CSS, DESeq2 variance-stabilising transformation).

Response: Sequencing depth ranged from 45,423 to 308,901 sequences; the median values were 113,211 (Q1= 71,588; Q3= 204,054). Given the high sequencing depth, the samples were not rarefied to preserve rare sequence data in the analyses. (Now highlighted in lines 158-161)

  • McMurdie PJ, Holmes S. Waste not, want not: why rarefying microbiome data is inadmissible. PLoS Comput Biol. 2014 Apr 3;10(4):e1003531. doi: 1371/journal.pcbi.1003531. PMID: 24699258; PMCID: PMC3974642.

This reference has now been incorporated into the manuscript (Reference 32 - (highlighted in line 160)

Conclusion

This manuscript presents novel, clinically relevant observations linking gut dysbiosis to ocular manifestations of SJS and SjD. The study addresses a significant gap in understanding the gut–eye–immune connection. After addressing the major methodological concerns (power, multiple testing, and confounders) and softening causal phrasing, the work will be suitable for publication.

Recommendation: Minor Revision

Reviewer 2 Report

Comments and Suggestions for Authors
  • The manuscript contains several conflicting statements regarding the number of enrolled and analyzed participants, leading to confusion:

-Abstract: reports 10 SJS, 10 SjD, and 10 healthy controls analyzed.

-Methods: states 10 SJS, 10 SjD, and 12 healthy controls included.  Also states that 10 controls were matched to SJS and 10 to SjD, with 8 overlapping and 2 unique individuals. Additionally, 1 SJS participant was excluded after sequencing, which would reduce the SJS microbiome dataset to 9. These descriptions are inconsistent and should be corrected for clarity and accuracy. The authors should provide a single, transparent accounting of participant enrollment and the final numbers used for microbiome and clinical analyses. The Abstract and all tables/figures must be updated accordingly to ensure complete consistency throughout the manuscript.

I recommend adding a brief flow description (or flow diagram) summarizing the number of recruited subjects per group, any exclusions, and reasons, and the final number analyzed per group for each comparison. This clarification is essential for the reader to interpret the study design and statistical validity accurately.

  • The study includes only 9 SJS patients and 10 SjD patients. This is a very small cohort, especially for a rare disease like SJS. Such a small sample size drastically reduces the statistical power of the study and makes the findings highly susceptible to influence by individual variations in gut flora. The correlations found may not be robust or reproducible in a larger population.
  • The manuscript does not include a statistical justification for the selected sample size. A power calculation or clear rationale is needed to determine whether the study is adequately powered to detect biologically meaningful differences in microbiome composition and clinical correlations. If this were intended as an exploratory pilot study, the authors should explicitly state this.
  • In the exclusion criteria, the authors state: “Exclusion criteria included … antibiotic, probiotic, or prebiotic use. The term “prebiotic use” is unclear. Prebiotics are typically dietary fibers consumed through normal eating habits. The Methods section does not describe any dietary assessment allowing identification of prebiotic intake. Please clarify whether this refers specifically to prebiotic supplements. If no dietary data were collected, this exclusion criterion should be revised or removed, and dietary habits should be acknowledged as an uncontrolled confounding factor (diet is one of the strongest modulators of the gut microbiome).
  • Beyond excluding recent antibiotic use, the manuscript does not provide sufficient information on systemic medication exposure. Apart from reporting that all SjD patients were taking hydroxychloroquine, no other systemic medications were documented for any group. Given that both SJS and SjD patients may be using drugs that can significantly impact gut microbiota, a complete medication history is essential to properly interpret microbiome differences. Please provide a detailed medication profile for all participants.
  • The manuscript notes that all SjD patients were under oral hydroxychloroquine treatment, but the potential influence of this medication on gut microbiota is not discussed. Hydroxychloroquine is known to modulate the intestinal microbiome, including reducing overall microbial diversity and altering the abundance of key commensal taxa in autoimmune disease. Therefore, the lack of difference between the SjD group and healthy controls in this study may be at least partially treatment-related, rather than indicating preserved microbiome composition in SjD. The authors should:

- Explicitly acknowledge hydroxychloroquine use as a major confounding variable in the Limitations section.

- Discuss how this medication could affect the interpretation of microbiome findings in SjD.

-Clarify whether treatment duration or dosage was collected and whether variability could influence the results.

Without addressing this point, the conclusion that SjD microbiota are “similar to healthy controls” may be misleading.

  • The single-center design means the results may not be generalizable to other populations with different genetic backgrounds, diets, and environmental exposures.
  • There are no citations in the Statistical Analysis and Sample Collection section.
  • I recommend updating the phylum name "Firmicutes" to its revised form, "Bacillota," to align with the other updated phylum names you correctly used, such as Bacteroidota and Actinobacteriota, creating internal consistency throughout the text, tables, and figures.

Author Response

POINT BY POINT

REVIEWER 2

Comments and Suggestions for Authors

  • The manuscript contains several conflicting statements regarding the number of enrolled and analyzed participants, leading to confusion:

-Abstract: reports 10 SJS, 10 SjD, and 10 healthy controls analyzed.

-Methods: states 10 SJS, 10 SjD, and 12 healthy controls included.  Also states that 10 controls were matched to SJS and 10 to SjD, with 8 overlapping and 2 unique individuals. Additionally, 1 SJS participant was excluded after sequencing, which would reduce the SJS microbiome dataset to 9. These descriptions are inconsistent and should be corrected for clarity and accuracy. The authors should provide a single, transparent accounting of participant enrollment and the final numbers used for microbiome and clinical analyses. The Abstract and all tables/figures must be updated accordingly to ensure complete consistency throughout the manuscript.

I recommend adding a brief flow description (or flow diagram) summarizing the number of recruited subjects per group, any exclusions, and reasons, and the final number analyzed per group for each comparison. This clarification is essential for the reader to interpret the study design and statistical validity accurately.

Response: Thank you for this relevant comment. We have thoroughly revised the manuscript to ensure complete clarity and consistency regarding the number of enrolled and analyzed participants. The description of participant numbers has now been fully harmonized across the Abstract, Materials and Methods, and Table 2 (summary of the demographic characteristics). We now clearly state that the study included 10 SJS patients matched to 10 healthy controls, and 10 SjD patients matched to a distinct group of 10 healthy controls. Additionally, one SJS participant was excluded after sequencing, resulting in a final dataset of 9 SJS patients for microbiome analyses. (Highlighted in lines 27-29, 87-92, and in legend of table 2)

All sections of the manuscript have been updated to reflect this structure, eliminating previous inconsistencies. Although a flow diagram was not added, we substantially improved the textual description. We enhanced the description and legend of Table 2, which now provides a clear and transparent accounting of recruitment, exclusions, and final sample sizes for each comparison. Given these revisions, an additional table is not necessary, as the updated Table 2 already conveys this information with sufficient clarity to ensure accurate interpretation of the study design and statistical validity.

-The study includes only 9 SJS patients and 10 SjD patients. This is a very small cohort, especially for a rare disease like SJS. Such a small sample size drastically reduces the statistical power of the study and makes the findings highly susceptible to influence by individual variations in gut flora. The correlations found may not be robust or reproducible in a larger population.

-The manuscript does not include a statistical justification for the selected sample size. A power calculation or clear rationale is needed to determine whether the study is adequately powered to detect biologically meaningful differences in microbiome composition and clinical correlations. If this were intended as an exploratory pilot study, the authors should explicitly state this.

Response: We fully agree with the reviewer that the small sample size, particularly in the SJS group, represents an important limitation. SJS is an extremely rare condition, which inherently restricts patient recruitment even in large tertiary centers. Moreover, this is the first study in the literature to investigate the intestinal microbiome in this patient population, further underscoring the challenges in assembling larger cohorts. For these reasons, and as now clearly stated in the manuscript, our study was designed as an exploratory pilot investigation, aimed at generating hypotheses rather than providing confirmatory statistical inferences.

This approach is consistent with prior gut-microbiome studies in Sjögren’s disease that used similarly small sample sizes without reporting power calculations, including Mendez et al. (n=13 SjD) and Moon et al. (n=10 SjD). These widely cited studies also emphasize the exploratory nature of microbiome characterization in autoimmune conditions.

We have now further emphasized throughout the manuscript that the rarity of SJS limits sample size and that the findings should be interpreted as exploratory rather than confirmatory, in accordance with the reviewer’s recommendation. (highlighted in lines 25, 83, 358 -360)

  • Mendez R, Watane A, Farhangi M, Cavuoto KM, Leith T, Budree S, et al. Gut microbial dysbiosis in individuals with Sjögren's syndrome. Microb Cell Fact. 2020 Apr 15;19(1):90. doi: 10.1186/s12934-020-01348-7. PMID: 32293464; PMCID: PMC7158097.
  • Moon J, Choi SH, Yoon CH, Kim MK. Gut dysbiosis is prevailing in Sjögren's syndrome and is related to dry eye severity. PLoS One. 2020 Feb 14;15(2):e0229029. doi: 10.1371/journal.pone.0229029. PMID: 32059038; PMCID: PMC7021297.

-In the exclusion criteria, the authors state: “Exclusion criteria included … antibiotic, probiotic, or prebiotic use. The term “prebiotic use” is unclear. Prebiotics are typically dietary fibers consumed through normal eating habits. The Methods section does not describe any dietary assessment allowing identification of prebiotic intake. Please clarify whether this refers specifically to prebiotic supplements. If no dietary data were collected, this exclusion criterion should be revised or removed, and dietary habits should be acknowledged as an uncontrolled confounding factor (diet is one of the strongest modulators of the gut microbiome).

Response: Thank you for this helpful clarification. The term “prebiotic use” has now been removed from the exclusion criteria to avoid ambiguity (highlighted in line 95), since we did not assess dietary intake or prebiotic-containing food consumption. As recommended, we have added a statement in the Limitations section acknowledging that dietary habits were not evaluated and represent an uncontrolled confounding factor, given their well-known influence on gut microbiome composition (highlighted in 360 -362).

-Beyond excluding recent antibiotic use, the manuscript does not provide sufficient information on systemic medication exposure. Apart from reporting that all SjD patients were taking hydroxychloroquine, no other systemic medications were documented for any group. Given that both SJS and SjD patients may be using drugs that can significantly impact gut microbiota, a complete medication history is essential to properly interpret microbiome differences. Please provide a detailed medication profile for all participants.

Response: Thank you for this important comment. In accordance with the reviewer’s suggestion, we have now added the Supplementary Table 1 summarizing all concomitant systemic and topical medications used by participants in both disease groups. (Supplementary Table 1).

-The manuscript notes that all SjD patients were under oral hydroxychloroquine treatment, but the potential influence of this medication on gut microbiota is not discussed. Hydroxychloroquine is known to modulate the intestinal microbiome, including reducing overall microbial diversity and altering the abundance of key commensal taxa in autoimmune disease. Therefore, the lack of difference between the SjD group and healthy controls in this study may be at least partially treatment-related, rather than indicating preserved microbiome composition in SjD. The authors should:

- Explicitly acknowledge hydroxychloroquine use as a major confounding variable in the Limitations section.

- Discuss how this medication could affect the interpretation of microbiome findings in SjD.

-Clarify whether treatment duration or dosage was collected and whether variability could influence the results. Without addressing this point, the conclusion that SjD microbiota are “similar to healthy controls” may be misleading.

Response: Thank you for this important comment. The duration of hydroxychloroquine (HCQ) use for each SjD patient is now clearly reported in the Supplementary Table 1, showing that all patients had been on HCQ 400mg/day treatment for two years or longer. As recommended, we have explicitly acknowledged hydroxychloroquine as a major potential confounding factor in the Limitations section, given its known effects on gut microbial diversity and composition.

We have also expanded the Discussion to address how HCQ use may influence the interpretation of microbiome findings in the SjD group and potentially attenuate differences relative to healthy controls. In addition, a paragraph was added discussing the possible confounding impact of topical and systemic anti-inflammatory medications used by patients in both disease groups. These revisions are highlighted in lines 340–357.

- The single-center design means the results may not be generalizable to other populations with different genetic backgrounds, diets, and environmental exposures.

Response: Thank you for the observation. The limitations of a single-center design and its potential impact on generalizability have now been explicitly acknowledged in the Limitations section (highlighted in lines 365-367).

- There are no citations in the Statistical Analysis and Sample Collection section.

Response: We appreciate this comment. References have now been added to both the Statistical Analysis (reference 33, line 168) and Sample Collection sections (reference 29, line 143) to support the methodologies employed. These additions provide clearer methodological grounding and address the reviewer’s concern.

- I recommend updating the phylum name "Firmicutes" to its revised form, "Bacillota," to align with the other updated phylum names you correctly used, such as Bacteroidota and Actinobacteriota, creating internal consistency throughout the text, tables, and figures.

Response: Thank you for noting this nomenclature inconsistency. The entire manuscript has been reviewed, and the updated phylum names have now been standardized throughout the text, tables, and Figures 2 and 3, ensuring complete internal consistency.

Round 2

Reviewer 2 Report

Comments and Suggestions for Authors

Dear authors,

I have reviewed the changes made, and I am very pleased to confirm that you have successfully incorporated my previous recommendations.  The revisions throughout the text are thoughtful and have enhanced the clarity and impact of the manuscript.